# *K*-mer Frequency Encoding Model for Cable Defect Identification: A Combination of Non-Destructive Testing Approach with Artificial Intelligence

Brijesh Patel [1], Zih Fong Huang [1], Chih-Ho Yeh [1], Yen-Ru Shih [1] and Po Ting Lin [1,2,*]

1 Department of Mechanical Engineering, National Taiwan University of Science and Technology, Taipei 10607, Taiwan; aero.brijesh@gmail.com (B.P.); z0955900208@gmail.com (Z.F.H.); jeff0918ma@gmail.com (C.-H.Y.); a0309cindy@gmail.com (Y.-R.S.)
2 Intelligent Manufacturing Innovation Center (IMIC), National Taiwan University of Science and Technology, Taipei 10607, Taiwan
* Correspondence: potinglin@mail.ntust.edu.tw

**Abstract:** This paper describes a non-destructive detection method for identifying cable defects using *K*-mer frequency encoding. The detection methodology combines magnetic leakage detection equipment with artificial intelligence for precise identification. The cable defect identification process includes cable signal acquisition, *K*-mer frequency encoding, and artificial intelligence-based identification. A magnetic leakage detection device detects signals via sensors and records their corresponding positions to obtain cable signals. The *K*-mer frequency encoding method consists of several steps, including cable signal normalization, the establishment of *K*-mer frequency encoding, repeated sampling of cable signals, and conversion for comparison to derive the *K*-mer frequency. The *K*-mer frequency coding method has advantages in data processing and repeated sampling. In the identification step of the artificial intelligence identification model, an autoencoder model is used as the algorithm, and the *K*-mer frequency coding method is used to introduce artificial parameters. Proper adjustments of these parameters are required for optimal cable defect identification performance in various applications and usage scenarios. Experiment results show that the proposed *K*-mer frequency encoding method is effective, with a cable identification accuracy rate of 91% achieved through repeated sampling.

**Keywords:** *K*-mer frequency encoding; cable defects; non-destructive testing; artificial intelligence



## 1. Introduction

The transition of traditional industries to smart manufacturing, as conceived by the Industry 4.0 paradigm, has created opportunities to utilize recent advances in artificial intelligence (AI) for increased productivity and safety [1–3]. Despite these advances, handling heavy payloads remains a major safety concern in a variety of manufacturing environments. Cranes are essential in heavy load handling since they can move objects weighing several metric tons. Cranes are designed to lift, lower, and transport heavy loads precisely and efficiently, facilitating the smooth operation of various industrial processes. Handling heavy loads, on the other hand, involves inherent risks and challenges [4]. Crane lifting ropes are subjected to significant stress and wear over time. The load weight, frequency of use, environmental conditions, and operational demands can all contribute to lifting cable deterioration [5]. It is critical to regularly monitor and inspect these ropes to ensure structural integrity and to avoid potential accidents or equipment failures. Lifting cables are particularly prone to failure and must be inspected and replaced regularly. Manual inspection procedures, on the other hand, are labor-intensive, time-consuming, subjective, and frequently necessitate the temporary halt of production processes [6].

Lifting cables are generally manufactured from steel wires, and may sometimes use synthetic fibers [7]. Steel wire cables are susceptible to damage during use, which can compromise their strength and pose safety risks. A comprehensive understanding of the various types of damage in steel wire cables is crucial for accurate assessment. These damage types include wire breakage, wear, deformation, rust, and fatigue. Fatigue, in particular, presents multiple manifestations, such as cracks, wire breakages, and slack. Proper assessment and recognition of these damage types are critical for steel wire cable integrity and safety [8]. Cable inspection and testing are critical safety measures carried out during the final stages of production and pre-installation operations [9]. Detecting and repairing defects in cables and wires are critical for avoiding financial losses and ensuring user safety and well-being. As a result, cable manufacturers and end users recognize the critical importance of performing multiple tests and inspections to mitigate potential risks and ensure reliable cable performance.

Steel cable fault detection methods are commonly classified as non-destructive or destructive. Non-destructive detection methods use specialized detection instruments or rely on the observation of abnormal phenomena to visualize defects or anomalies within the cable material [10]. The captured signals, images, or parameters are then subjected to extensive analysis, evaluation, and judgment. Non-destructive testing is distinguished by its ability to assess the internal condition of a structural material without causing any damage to its structure, performance, or dimensional integrity. This ensures that the original shape and functionality of the cable are preserved throughout the testing process. Metal cable detection methods include manual detection, automated optical detection (AOI), and magnetic leakage detection [11]. Magnetic flux leakage detection evaluates the condition of metal cables by measuring the loss of metallic area (LMA). Relevant guidelines such as ISO 4309 [12] are followed to ensure compliance with industry standards. This standard covers the fundamental principles of detection, procedures, appropriate instruments, and calibration methods required for accurate and reliable cable testing. Following these established standards is critical for maintaining consistency and quality in the detection process. The magnetic properties of the materials, magnetization levels, and defect characteristics such as depth, width, length, inclination angle, and others all influence the effectiveness of magnetic leakage detection equipment [13].

Detecting faults in metal cable systems, essential for safety and reliability in applications like bridges and elevators, faces complex testing environments with signal interference and noise. After signal acquisition and processing, signal analysis becomes highly complex due to decreased signal-to-noise ratio and inspection sensitivity. Various signal processing techniques, including impulse filters, data fusion, and feature extraction, have been explored to address these challenges [14,15]. Incorporating advanced mathematical techniques into physical testing for various inspection methods is a catalyst for precision and efficiency [16]. This involves employing Fourier analysis, machine learning algorithms, image reconstruction, and simulations to enhance inspection processes [17,18]. These techniques unlock solutions to intricate challenges, propelling the field of inspection technology forward. Innovative algorithms, driven by supervised and unsupervised learning methods and neural network techniques [19–23], have emerged to meet evolving engineering requirements. For instance, Khatir et al. introduced a novel PSO-YUKI algorithm with RBF for rapid damage identification in CFRP laminates, outperforming traditional PSO in double crack depth assessment. Qinghua Mao et al. [24] introduced an improved decision tree support vector machine (SVM) algorithm, validated through experiments, to enhance the classification accuracy of metal cord conveyor belt defects. The implementation of an artificial neural network (ANN) alongside advanced algorithms for structural health monitoring of laminated composite plates has a primary focus on damage localization and quantification [25]. Additionally, an ANN is employed to predict the displacement of composite pipes subjected to varying velocities. This reflects the continuous advancement in the field of ANN techniques for structural health monitoring [26]. Beyond traditional machine learning, deep learning techniques [27], particularly convolutional neural networks

(CNNs) [28], have gained prominence in various domains, including signal recognition. Pyakillya et al. [29] employed a fully connected neural network (fully convolutional networks, FCN) and a one-dimensional convolutional neural network (1D CNN) to analyze and classify ECG signals, while Zhang and Wu [12] applied deep belief nets (DBN) to analyze sound signals and voice activity detection. Gongbo Zhou et al. [30] demonstrated the utility of CNNs for real-time fault detection in the context of metal cable fault detection. At the same time, Zhiliang Liu et al. [31] developed CNN-based methodologies for surface defect detection, showcasing their powerful learning capabilities and high diagnostic accuracy in metal cable systems. Liu et al. [32] examined wire rope defect recognition using MFL signal analysis and 1D CNNs, emphasizing signal processing and machine learning's role in enhancing accuracy, reflecting a current trend in the field. These examples highlight the crucial role of data processing in achieving effective signal identification and defect detection in metal cable systems.

*K*-mers serve versatile roles in bioinformatics, encompassing quality control for sequence generation, metagenomic classification, and genome size estimation [33,34]. Maplason et al. described KAT as a tool for NGS data quality control and *K*-mer analysis, providing k-value guidance and tool comparisons [35]. Breitwieser et al. presented KrakenUniq, which melds Kraken's speed with efficient unique *K*-mer coverage assessment in metagenomics [33]. Taha et al. extracted amino acid *K*-mer features for machine learning-based quantitative antimicrobial resistance (AMR) prediction and offered model interpretation for biological insights [36]. Machine learning methods based on *K*-mers have also shown promise in pattern recognition problems, achieved by quantifying fixed-length *K*-mer frequencies in DNA sequences [37,38]. Akkaya et al. used deep learning to evaluate sequence representations and emphasize their role in model performance. They showed that preserving *K*-mer relationships is crucial for better results [39]. Integrating *K*-mer frequencies with deep learning methods enhances data classification accuracy, leveraging their complementary strengths.

In this study, artificial techniques are employed for the training and identification of signal data. The signal obtained through the magnetic leakage detection equipment undergoes normalization. Subsequently, the proposed *K*-mer frequency encoding method is applied. Finally, the autoencoder (AE) training recognition is the reason for choosing an autoencoder because it can identify non-defective data and defective data only by inputting non-defective data.

## 2. Material and Methods

### 2.1. Metal Cable Fault Detection

Steel metal cables often have local flaw defects, such as wear, broken wires, extrusion, thickness variations, corrosion, and kinks. Traditional inspection methods for steel cables typically involve visual inspection or size measurements, which are prone to human error. Image processing is used in precision measurement techniques such as AOI to detect external defects in metal cables [40]; however, these methods have inherent limitations when it comes to identifying internal flaws. Such internal defects can give rise to hidden internal stress and often elude detection through visual examination alone. Consequently, the need for non-destructive testing techniques like magnetic flux leakage (MFL) becomes increasingly imperative, as they offer the ability to probe beneath the surface and unveil concealed flaws within these vital components, enhancing both safety and reliability [41].

2.1.1. Metal Cable Fault Detection Using Magnetic Flux Leakage

The magnetic flux leakage (MFL) method, a non-destructive testing technique for evaluating the integrity of ferromagnetic materials, takes center stage in this study. With a primary focus on steel cables, this method leverages the fundamental principles of magnetism and sophisticated sensor technology to identify flaws and defects in these critical components. As we delve deeper into this exploration, it becomes evident that advancing cable inspection and maintenance techniques is not just a matter of protocol but

a pivotal endeavor driven by public safety and economic stability considerations. The MFL method is represented in a three-dimensional (3D) view in Figure 1.

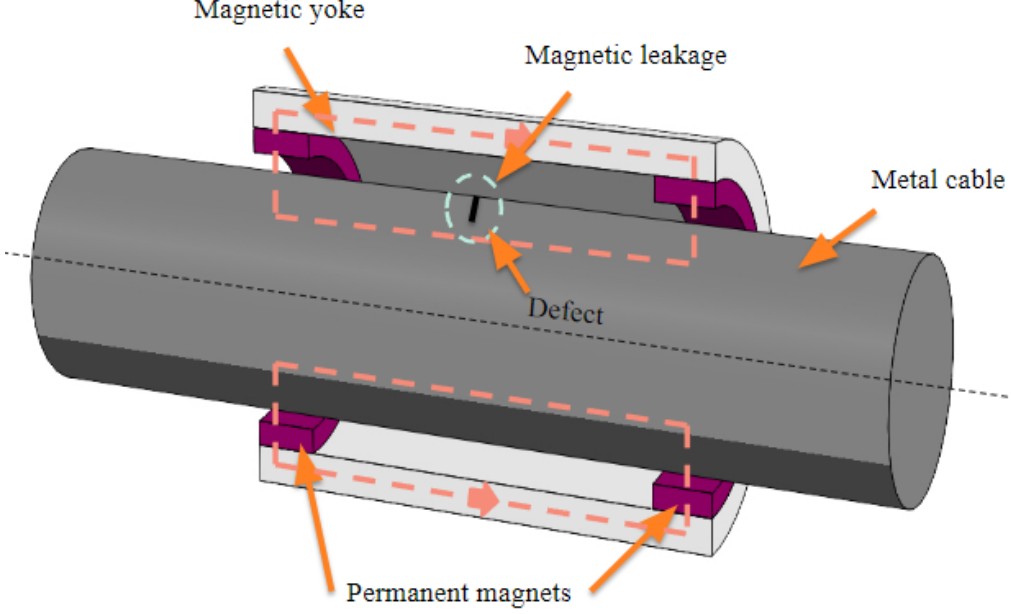

**Figure 1.** Representation of magnetic flux leakage method.

This study employs a magnetic flux leakage detection method for metal cable inspection. The schematic representation of the principles underlying the typical magnetic flux leakage (MFL) method is shown in Figure 2. The metal cable is subjected to the influence of a powerful permanent magnet or electromagnet. Due to the high magnetic permeability of the metal cable material, the cable induces a magnetic field when it is in proximity to the magnet, resulting in magnetic saturation. Surface or internal flaws in the steel cable disrupt and distort the magnetic field, leading to the leakage of magnetic flux into and out of the cable surface. This phenomenon is detected using sensors, which record the relative position and intensity of the flaw signal, facilitating the detection of steel cable defects.

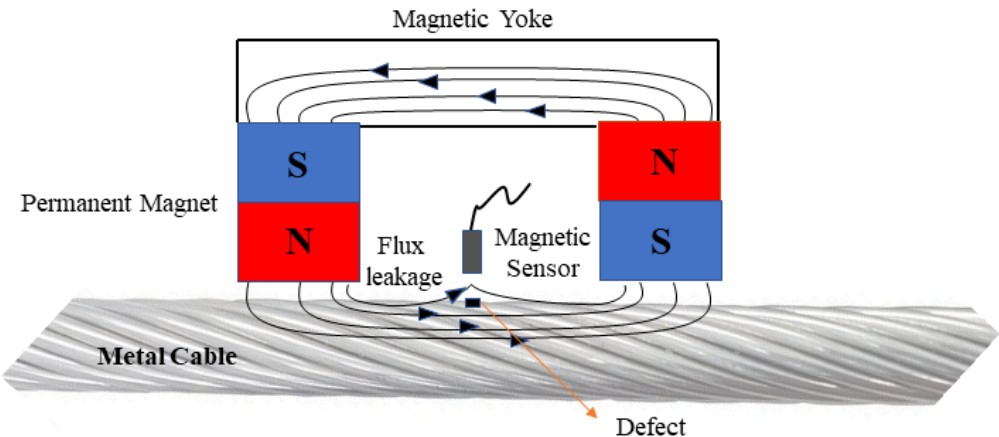

**Figure 2.** Schematic representation of the principles of the magnetic flux leakage (MFL) method.

2.1.2. Metal Cable Fault Data Acquisition Using Magnetic Flux Leakage Equipment

In this study, we aimed to comprehensively assess the performance of magnetic flux leakage equipment specifically designed for detecting metal cable defects, as shown in Figure 3. To ensure the reliability and robustness of our evaluations, we meticulously controlled various environmental parameters. This study employs compacted steel wires with dimensions measuring 26 mm composed in a 6 × 36 configuration and having an

overall length of 3 m. This research employed two distinct types of steel wires. The first type comprises wires without any irregularities or defects, while the second type includes wires with irregularities, encompassing cutting damage, signs of wear and tear, and instances of external wire breakage damage, all of which manifest along the entire length of the wire. Our experiments were conducted within a temperature range spanning from −20 °C to 40 °C, mirroring the cable's potential operating conditions. Atmospheric pressure was rigorously monitored and maintained within the range of 86 kPa to 206 kPa, and humidity levels were carefully controlled, not exceeding 85%, to mitigate potential interference due to moisture. Additionally, we took great care to create an environment free from strong vibrations, dust, strong magnetic fields, and corrosive substances, which could adversely affect the equipment's performance. The equipment was supplied with a stable 5 V power source throughout the experiments to ensure uninterrupted operation. Our evaluation encompassed a broad spectrum of cable sizes, ranging from wire diameters of 1.5 mm to 300 mm, and we investigated the equipment's performance across varying cable speeds, with particular emphasis on its optimal performance at 1.0 m/s. Our testing included assessments of the equipment's low frequency (LF) testing capabilities, achieving a remarkable broken wire detection accuracy of 98.9%. Additionally, we scrutinized its low magnetic attraction (LMA) testing capabilities, with an allowable error of ±0.055% and an accuracy value error of ±0.2%. Furthermore, we evaluated the equipment's distance testing capability, maintaining an allowable error of ±0.3%. To ensure the equipment's consistent performance, we conducted 100 tests and verified that it consistently achieved an accuracy of over 98.9% for broken wire detection. Throughout the experiments, we closely monitored environmental conditions, meticulously calibrated equipment settings, and rigorously documented test results to comprehensively assess the magnetic flux leakage equipment's performance under real-world operating conditions.

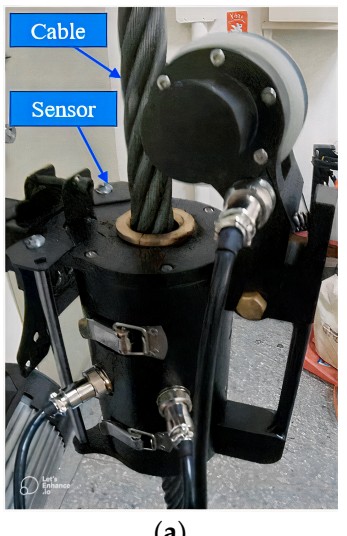
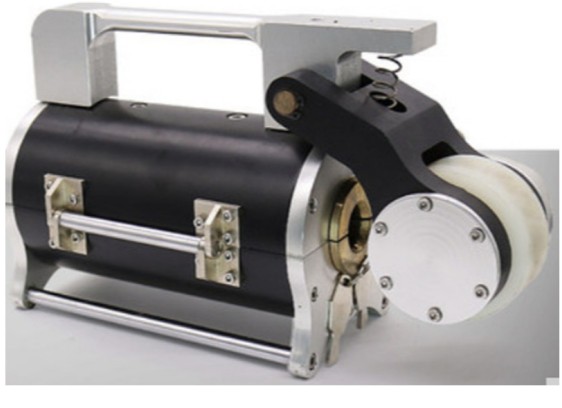

(**a**)          (**b**)

**Figure 3.** Defect detection equipment for metal cables based on MFL. (**a**) Experimental arrangement of the equipment; (**b**) MFL-based defect detection equipment.

The experimental setup for the complete signal acquisition process showing the sequential steps involved in capturing the cable signal is illustrated in Figure 4, where the metal cable being tested is placed within the magnetic leakage detection equipment, which records the cable signal at 10 times per second sampling rate. A digital converter converts the analog signal into a digital signal. The digitalized signal is then sent to a computer for further analysis and processing. Also, Figure 5 shows the actual signal obtained by the magnetic flux leakage detection equipment, providing a visual representation of the data acquired. In the event of a cable defect, MFL detects it, resulting in a change in the signal frequency, as visually illustrated in Figure 6.

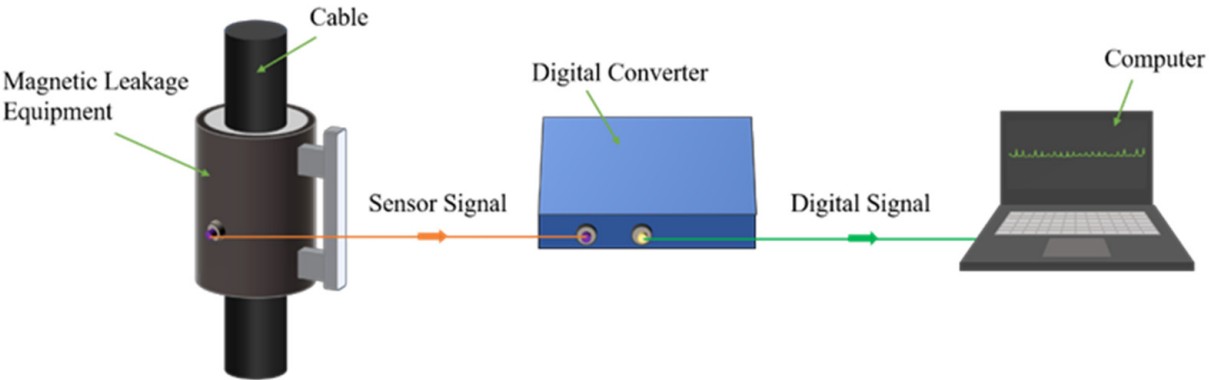

**Figure 4.** Experimental setup for the signal acquisition process.

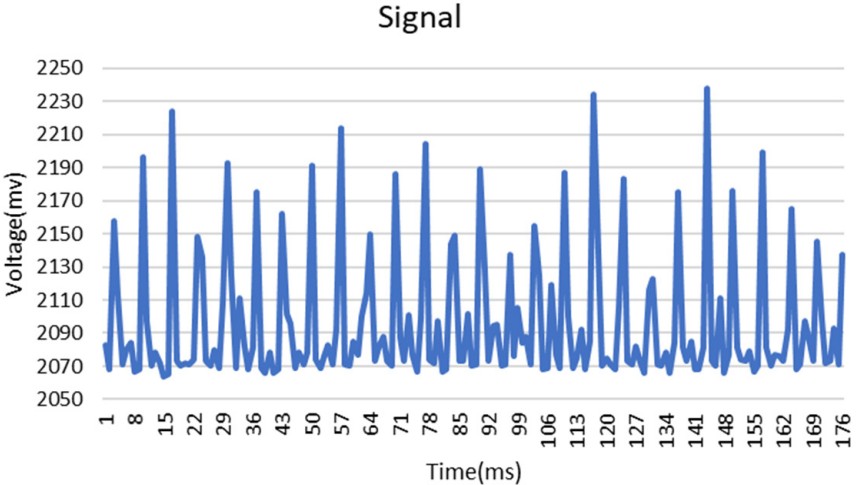

**Figure 5.** Visualization of signal from MFL equipment.

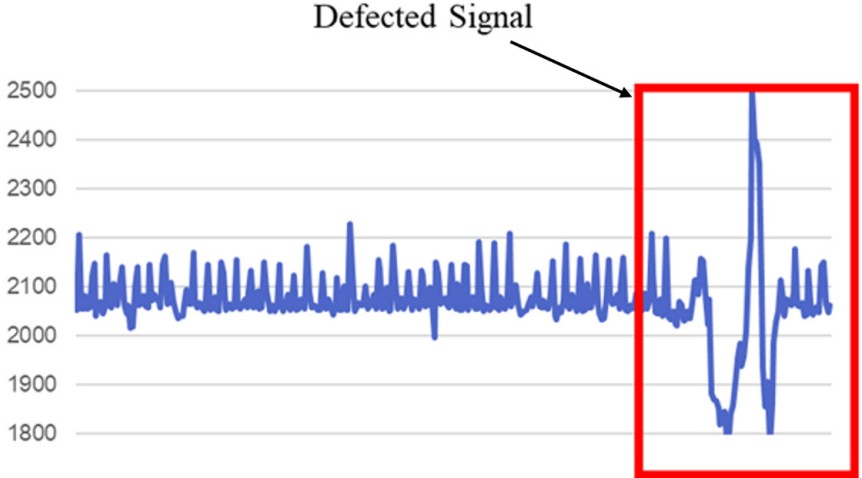

**Figure 6.** Visualization of defected signal from MFL equipment.

### 2.2. K-mer Frequency Encoding Method

*K*-mer encoding was originally applied in genetic testing methods to detect the genetic characteristics of two species, A and B. If the frequencies of detected genetic features are similar, both species are considered the same; otherwise, they are considered different species. Recently, this method has shown promising results in image processing and computer vision. Therefore, this paper utilizes this approach to process one-dimensional signals. The *K*-mer frequency coding method is proposed in this study for the training

process of one-dimensional data, as shown in Figure 7. The procedure is divided into two stages: the left side represents deep network learning training, and the right side represents the application of identifying cable defects. Prior to these stages, data pre-processing normalization was performed, as recommended by previous research [42]. Normalization technique is beneficial in scenarios where there is excessive noise, large gaps and accelerated convergence. It can enhance the model's convergence speed and overall accuracy. Min–max normalization is used in this study to scale the data into intervals of equal proportions for this purpose, with the calculation method outlined in Equation (1).

$$X_{nom} = \frac{X - X_{\min}}{X_{\max} - X_{\min}} \in [0, 1] \tag{1}$$

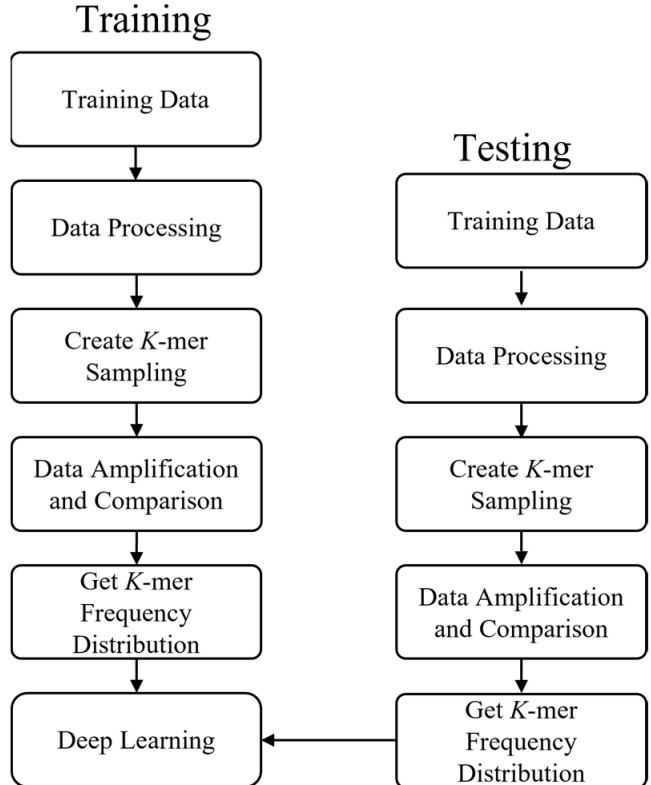

**Figure 7.** One-dimensional data training flowchart of *K*-mer frequency encoding method.

After normalizing the cable signal data, the *K*-mer frequency encoding method was applied. Figure 8 illustrates the operational procedure of *K*-mer frequency encoding method. In this method, normalized value are input, where M is the length of the normalized cable signal data and *K* is the user defined truncated data length. After defining the value of *K*, the data is divided according to this length. This action allows for repeated sampling of cable signal data. Binary conversion is performed after this division to improve data comparison and simplify the representation. The binarization threshold value is an artificial parameter that must be tailored to specific applications and usage scenarios. Following the binary conversion, the data is subjected to *K*-mer frequency calculation. The distribution pattern is determined by *K*, and the distribution style algorithm employs permutation and combination principles. When *K* equals 2, a total $2^2$ of permutations and combination are generated and then compared with the binarized signal data to calculate the number of permutations and combinations. Through *K*-mer frequency calculation, the distribution pattern of this data can be determined. The count resulting from the frequency calculation is input into the AutoEncoder model for training and identification.

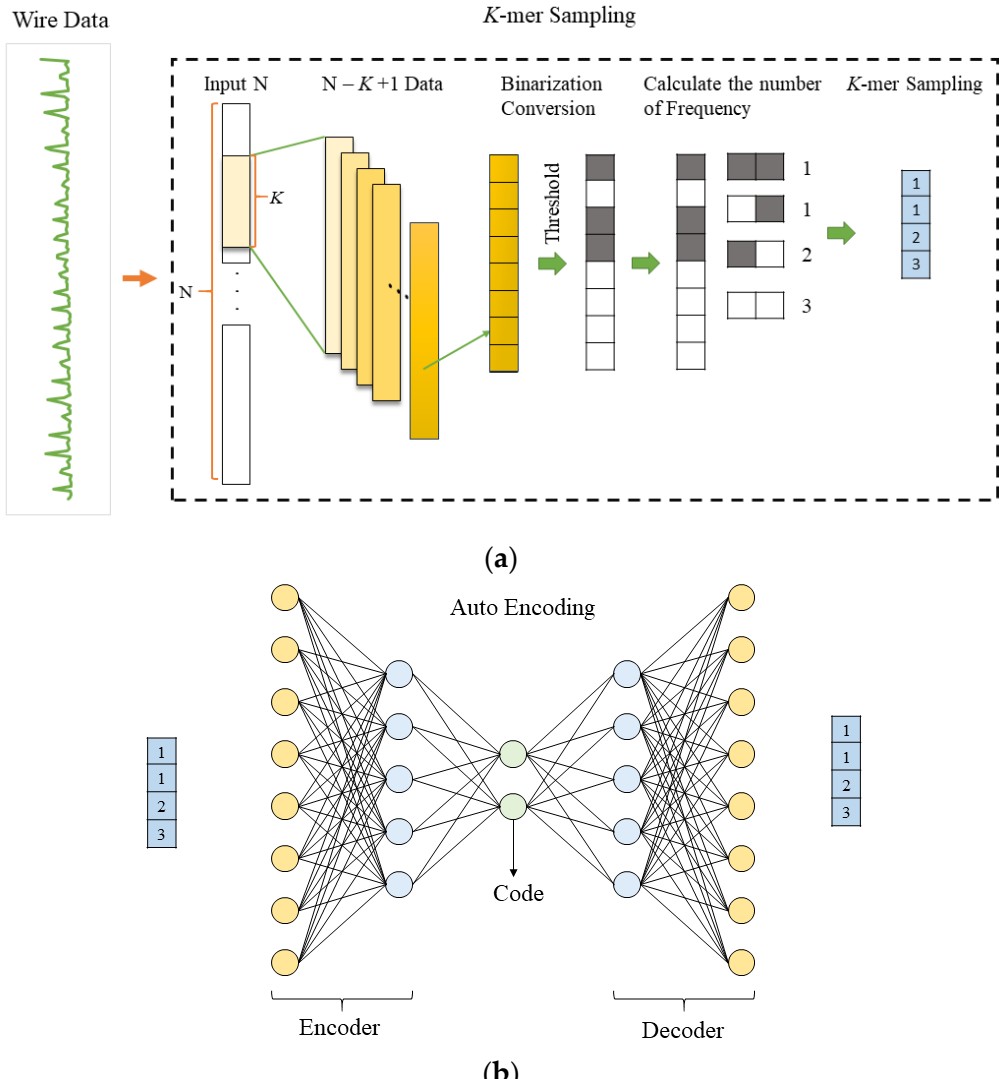

**Figure 8.** *K*-mer frequency encoding flowchart. (**a**) *K*-mer sampling process; (**b**) autoencoding process.

The deep learning approach employed in this study utilizes an autoencoder architecture [43,44]. This architecture facilitates tasks such as dimension reduction and feature extraction. It consists of two components: the Encoder, responsible for compression, and the Decoder, responsible for decompression. The compression and decompression processes aim to ensure that the output values align with the input values, as depicted in Figure 9. When inputting signal data, they undergo compression through the Encoding layers, resulting in reduced dimensionality and feature extraction. This process is repeated until they reach the Code layer, and subsequently, the decoding layers reconstruct the original signal data from the Code layer.

The Autoencoder model is typically used for data reconstruction. The model's goal is to generate results that closely match the input data by undergoing the process of encoding and decoding. During the training process, both the encoding and decoding layers are optimized, making them progressively more skilled at reconstructing the input data. Exploiting this characteristic, we developed an Autoencoder model that is exclusively trained with normal cable data for the purpose of data reconstruction. As a result, this model excels at reconstructing normal cable data, providing results that closely resemble the original data. However, when dealing with abnormal cable data, the model's ability to reconstruct is comparatively weaker due to a lack of training on how to reconstruct abnormal data. By comparing the differences between the data before and after reconstruction, we can

determine whether a cable has damaged sections. If the error value is excessively high, it is categorized as abnormal data.

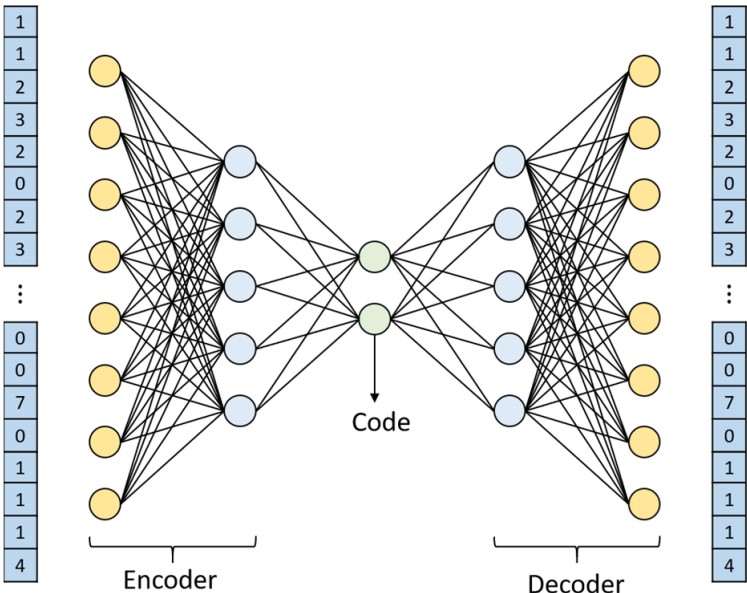

**Figure 9.** Autoencoder flowchart.

## 3. Results

In this study, the training data exclusively consisted of non-defective cable signals. During the testing phase, an equal distribution of 50% defective and 50% non-defective cable signals was employed to evaluate the model's performance. This deliberate choice enabled the autoencoder to learn the best method for restoring non-defective signals. The autoencoder is distinguished by its ability to effectively reconstruct the data on which it was trained. During the training phase, the reconstruction error associated with non-flawed signals is lower than the reconstruction error associated with flawed signals. As a result, in practice, the reconstruction error value can be used to determine whether or not a signal is flawed. The *K*-mer frequency encoding method was used to process the cable data accessed within the computer system. The resulting processed data were used to train the autoencoder model. Subsequent experiments were carried out to investigate the differences between training models with unprocessed data and training models with *K*-mer frequency encoded data. These experiments also aimed to identify the specific parameters under which *K*-mer frequency encoding produces superior results in cable signal analysis.

*3.1. Comparative Analysis of Cable Defect Identification Data Using K-mer Frequency Encoding Method*

In this study focused on cable defect identification, the experiment was conducted to compare the accuracy of a model trained using two different approaches: the *K*-mer frequency coding method and the use of original data. The experiment involved dividing the data and evaluating the accuracy of each approach. The accuracy of the model trained with the original data is presented in Figure 10, reaching a maximum of 81%. On the other hand, the accuracy of the model trained using the *K*-mer frequency coding method is displayed in Figure 11, achieving a maximum accuracy of 91%. These results indicate that employing the *K*-mer frequency encoding method effectively enhances the accuracy of the autoencoder model. Additionally, notable improvements were observed in each parameter.

*3.2. Comparative Analysis of Repeated Sampling Data*

The experimental study investigated the effects of repeated sampling data on a cable defect identification model. The total number of split data was determined by the value of *K*, with smaller *K* values resulting in a greater number of repeated sampling data. The

experiment considered three different *K* values to analyze this relationship: 10, 30, and 70. Figure 12 compares the accuracy rates of repeated sampling data, with blue representing one condition, orange representing another, and green representing a third. Based on the accuracy rates, it is clear that when the *K* value is smaller, the model's accuracy rate improves more noticeably.

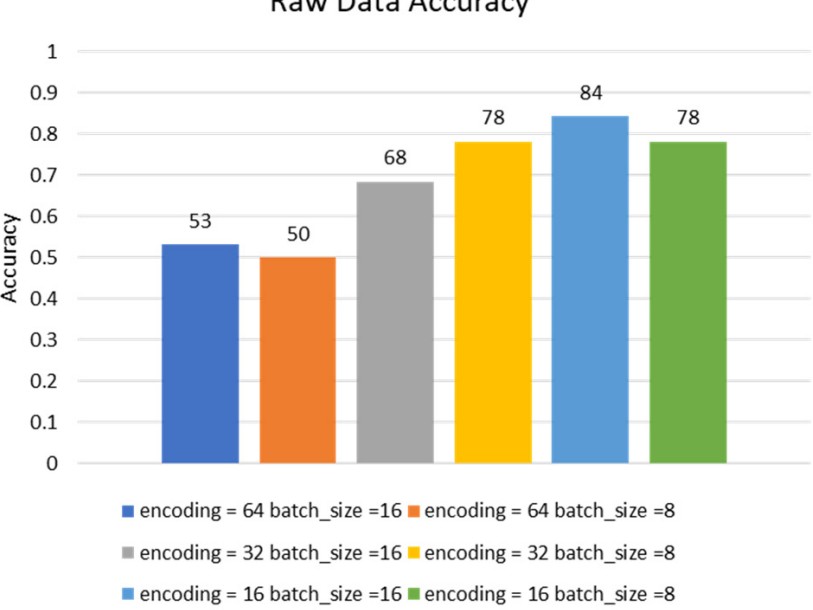

**Figure 10.** Raw data accuracy.

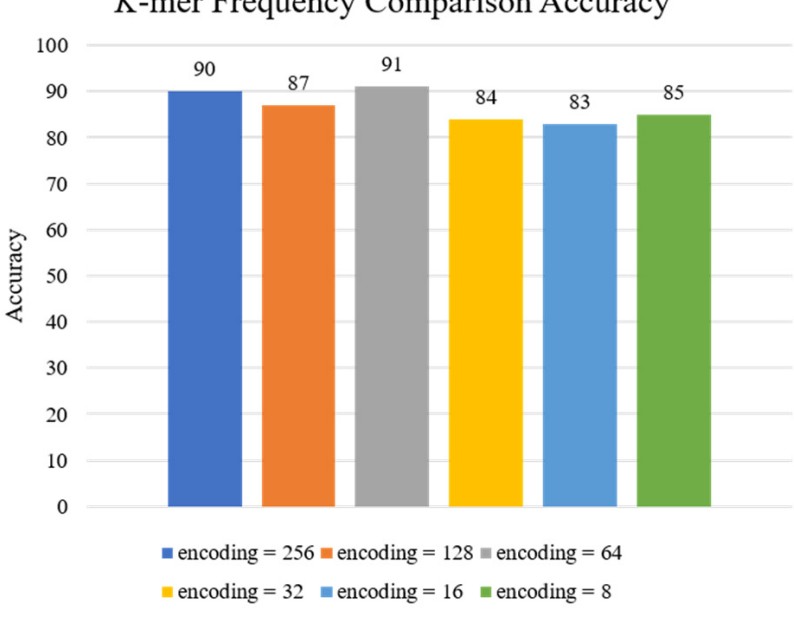

**Figure 11.** *K*-mer frequency comparison accuracy.

*3.3. Comparative Analysis of K-mer Frequency*

The experimental comparison factor explores the connection between *K*-mer frequency sampling and the $2^K$ identification model for sampling defects. Firstly, the frequency code was transformed into a numerical value for binary conversion, and then the occurrence count of the code was determined. As illustrated in Figure 13, from left to right, there are different coding arrangement styles: *K* = 4, *K* = 5, *K* = 6, *K* = 7, *K* = 8, *K* = 9. The model

clearly achieves the highest accuracy at 91%. Furthermore, as the value of *K* grows larger, the accuracy rate decreases.

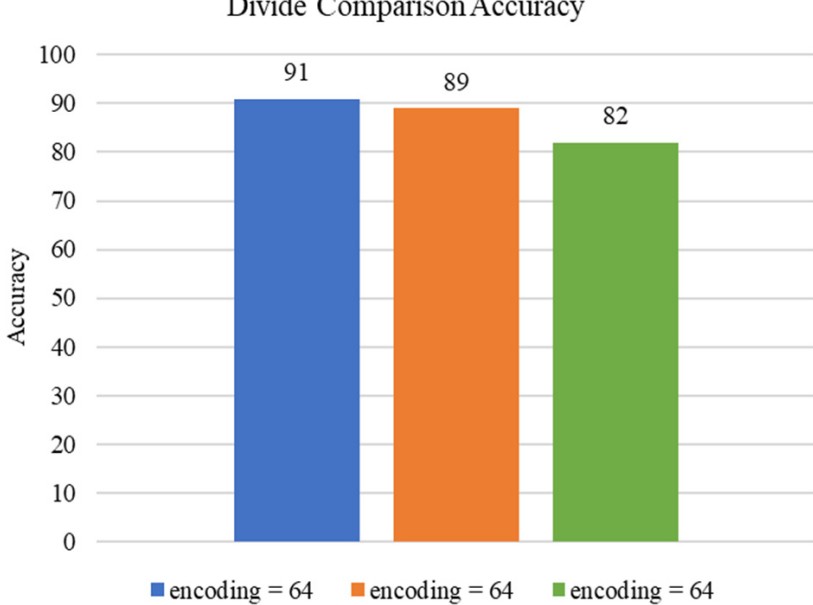

**Figure 12.** Divide comparison accuracy.

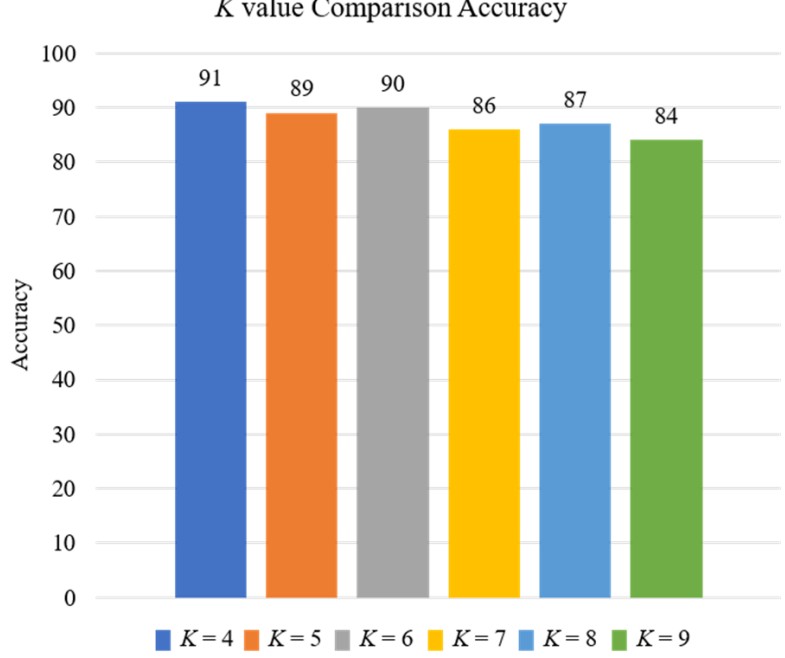

**Figure 13.** *K* value comparison accuracy.

### 3.4. Comparision of the K-mer Frequency Encoding Method with the Classical Method

Our analysis centered on the cable signal, which represents one-dimensional time series data, and we compared it with a dedicated 1D CNN tailored for processing 1D signal data; the accuracy results are illustrated in Figure 14. In both scenarios, *K*-mer sampling was a critical preprocessing step. It is noteworthy that the 1D CNN model achieved an accuracy rate of 82%, while the method of this study achieved a substantially higher accuracy of 91%. It is worth emphasizing that both models employed a *K* value of 4, ensuring consistency in this aspect of the comparison. This substantial improvement in accuracy demonstrates the effectiveness of our approach in handling one-dimensional time series data.

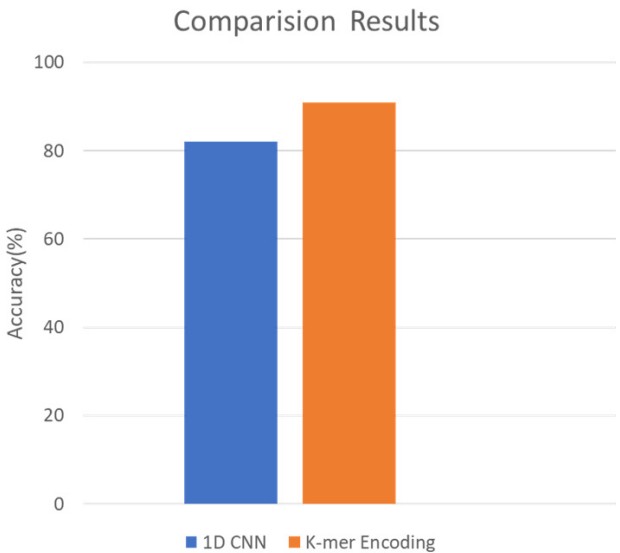

**Figure 14.** Accuracy comparison with the classical method.

## 4. Discussion and Limitations

This study has yielded significant insights into the effectiveness of artificial intelligence strategies for cable defect identification. We initiated our investigation by employing non-defective cable signals to train an autoencoder, a crucial choice that allowed the model to excel in reconstructing non-flawed signals, thereby establishing a practical approach to identify flawed cable signals based on reconstruction error values. This observation holds substantial practical implications, suggesting that reconstruction error values can serve as reliable indicators for detecting cable defects.

A notable contribution of this study was introducing the *K*-mer frequency encoding method for cable data processing. Our subsequent experiments aimed to evaluate its impact on the accuracy of the autoencoder model for cable defect identification. Our comparative analysis revealed compelling results, with the model trained using the *K*-mer frequency coding method achieving a remarkable maximum accuracy of 91%. This outcome significantly outperformed the model trained with original data, which reached a maximum accuracy of 81%. These findings underscore the substantial enhancement in accuracy that *K*-mer frequency encoding brings to cable signal analysis. Additionally, our research explored the effects of repeated sampling data on the cable defect identification model, showcasing a clear relationship between smaller *K* values and improved model accuracy. This suggests that the choice of *K* value plays a pivotal role in enhancing the model's ability to identify cable defects accurately. Furthermore, our comparative analysis of *K*-mer frequency sampling demonstrated that a *K* value of 4 yielded the highest accuracy at 91%, with accuracy declining as *K* values increased. This underscores the significance of selecting an appropriate *K* value when employing the *K*-mer frequency encoding method in cable signal analysis, offering promising avenues for optimizing cable defect identification in industrial applications.

One of the limitations of this study is that the method has been primarily applied to smaller datasets, and using it on larger datasets could result in generating an even larger amount of data, which may significantly extend the model's training time. This could pose challenges in real-world applications, where efficient training and inference times are crucial. Moreover, splitting the data too finely to accommodate larger datasets could lead to the loss of important data features, potentially affecting the model's performance. The model may suffer from decreased accuracy and generalization capabilities when faced with data scarcity due to excessive data splitting. Furthermore, the computational resources required for training and handling large datasets should be carefully considered to ensure efficient implementation in real-world applications. Despite these limitations, the proposed

method shows promising results on smaller datasets and lays the foundation for further exploration and optimization in the context of larger datasets and real-world scenarios.

## 5. Conclusions

In this study, a novel approach is proposed to address the limitations of current measurement equipment in the establishment of assessment standards for cable defect identification during magnetic leakage testing. Traditionally, assessment standards are set by the industry and may not consider the practical conditions encountered on-site, such as variations in cable picking angle, shape, and width. To overcome these challenges, the research introduces artificial intelligence (AI) as a powerful tool for establishing adaptable assessment standards that can dynamically adjust to the specific working conditions on-site. The utilization of an autoencoder model in this study revolutionizes the identification process of signals generated by magnetic leakage detection equipment. This model enables inspectors to define inspection standards tailored to conditions different from the on-site resources. The model's training only requires input data from defect-free cables with the specific conditions encountered on-site, eliminating the potential errors caused by variations between industry settings and real-world scenarios. By relying on AI to formulate assessment standards, the accuracy and reliability of cable defect identification are significantly enhanced.

Moreover, this study investigates the effectiveness of incorporating the $K$-mer frequency coding method into the model. The results reveal that the model with the coding method achieves an impressive accuracy rate of 91%, surpassing the accuracy rate of 81% obtained by the model without coding. This finding demonstrates the efficacy of the $K$-mer frequency coding method in improving the model's performance and reinforcing the accuracy of cable defect identification. This research showcases the immense potential of combining artificial intelligence identification with measurement equipment in cable defect identification. The proposed approach addresses the limitations of current assessment standards and opens up possibilities for further advancements. Future research directions may involve integrating image-based artificial intelligence identification techniques to enhance the precision of cable detection. Additionally, exploring synergies with other detection equipment can lead to formulating comprehensive inspection and assessment standards, empowering artificial intelligence to identify and categorize cable defects under diverse conditions adaptively.

**Author Contributions:** Conceptualization, P.T.L.; methodology, P.T.L.; software, Z.F.H. and C.-H.Y.; formal analysis, Z.F.H. and C.-H.Y.; investigation, C.-H.Y. and P.T.L.; writing—original draft preparation, B.P.; writing—review and editing, Y.-R.S. and B.P.; visualization, P.T.L. and B.P.; supervision, P.T.L.; funding acquisition, P.T.L. All authors have read and agreed to the published version of the manuscript.

**Funding:** The supports from the National Science and Technology Council, Taiwan (grant number NSTC 112-2218-E-011-009) and the Intelligent Manufacturing Innovation Center (IMIC), which is a Featured Areas Research Center in the Higher Education Sprout Project of the Ministry of Education (MOE), Taiwan were appreciated.

**Data Availability Statement:** The data presented in this study are available on request from the corresponding author. The data are not publicly available due to privacy.

**Conflicts of Interest:** The authors declare no conflict of interest.

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
