# Peer review of "K-mer Frequency Encoding Model for Cable Defect Identification: A Combination of Non-Destructive Testing Approach with Artificial Intelligence"

_inventions, doi:10.3390/inventions8060132_

Round 1

Reviewer 1 Report

This work focuses on the development of a non-destructive detection method using K-mer frequency encoding combined with magnetic leakage detection equipment. Artificial intelligence in cable defect identification by employing an autoencoder model and introducing artificial parameters through K-mer frequency coding. 

The authors need to address these points:

1. While the K-mer Frequency Encoding Method plays a significant role in this study, the authors have not sufficiently elaborated on its background or provided references related to this technique.

2. The authors need to provide the details related to the process of the AutoEncoder training and training results.

3. The defect detection experiment is not clear, figures 3 and 4 show the experiment setup, but there is no figure showing information related to the defect and its parameters, its size, position, type, or whether there are multiple damages or single damage.

4. What does encoding = 64 means? and there are no mentions of the batch size in the text.

5. The accuracy should also be compared to classical methods and machine learning methods such as the ones mentioned in the introduction. 

6. The authors should show awareness of advanced mathematical techniques in this field such as: "A hybrid multiphase flow model for the prediction of both low and high void fraction nucleate boiling regimes", "A new approach to improve ill-conditioned parabolic optimal control problem via time domain decomposition" and "Optimal Axial-Probe Design for Foucault-Current Tomography: A Global Optimization Approach Based on Linear Sampling Method"

7. The authors should also show awareness of existing approaches in damage detection using machine learning and optimization algorithms, such as: "A new hybrid PSO-YUKI for double cracks identification in CFRP cantilever beam"Enhanced ANN Predictive Model for Composite Pipes Subjected to Low-Velocity Impact Loads", "Damage assessment in laminated composite plates using modal Strain Energy and YUKI-ANN algorithm" and " Deep Neural Network and YUKI Algorithm for Inner Damage Characterization Based on Elastic Boundary Displacement".

8. remove this unrelated text 

"It is important to note that abbreviations and acronyms

242 should be defined when first mentioned in the text, even if they have already been defined

243 in the abstract. Abbreviations like IEEE, SI, MKS, CGS, sc, dc, and RMS do not require

244 further definition. Avoid using abbreviations in the title or headings unless necessary. "

no comment

Author Response

Dear Reviewer,

We are thankful to you for your valuable comments and suggestion. We have tried our best to provide the response to every query.

Thanks 

Reviewer 2 Report

The paper presents the study on the non-destructive inspection of metal cables (identification of cable defects) carried out under various controlled environmental parameters and for various sizes of these cables. The Authors make use of magnetic leakage detection method aided with artificial intelligence algorithms. It should be mentioned that the used detection technique is well known and widely used. However, the novel contribution deals with the approach of K-mer frequency encoding that seems interesting due to its confirmed efficiency in other fields. This approach is used to analyze one-dimensional signals. The constructed sensor provides interesting and useful data for validating the proposed defect detection approach. The Authors investigate deeply the properties of the applied K-mer frequency encoding method under its various settings. In the reviewer’s opinion, the work presents interesting results, however there are some issues that need to be addressed and more comprehensive comments should be added in the corrected version of the manuscript. Moreover, some doubts arise regarding the applicability of the presented approach to real cases of damaged wires.

What amount of data is necessary to tune (parameterize) the algorithm so that it would be sufficiently sensitive to detect the defects of the expected size and type in the inspected wires?

Row 81: “due to increased signal-to-noise ratio”  I believe it should be “due to the decreased signal-to-noise ratio”  which seems a challenge.

194: “The procedure is divided into two stages: the left side represents deep network learning training, and the right side represents the application of identifying cable defects.” – What are the mentioned left and right sides in the shown vertical scheme?

247: the Authors state  “The cable signals used in this study are all non-defective signals” and then „As a result, in practice, the reconstruction error value can be used to determine whether or not a signal is flawed” Are there provided the results for the defected wires to prove that statement?

The Authors focus on the reconstruction error values for the proposed approach and search for reliable defects indicators. However, the question arises: how to assure the claimed efficiency on defects detection analyzing the intact structure only?

278: “A formula determines the total number …’   What formula?

Minor issues and flaws:

Row 49, 53: “Steel” -> “steel”

55: “Detecting and correcting defects in cables and wires” -> “Detecting defects in cables and wires and their repair”

97: “Liu et al. [26]examine” -> “Liu et al. [26](space)examine”

115: “cables [27]. ,” -> “cables [27],”

207: “a comprehensive value is” -> “comprehensive data are”

209: “the value” -> “the entered value”

242-245- editorial error, the following text up to the end of the paragraph should be removed “It is important to note that abbreviations and acronyms should be ....”

Author Response

(The authors gave the same response as above.)

Round 2

Reviewer 1 Report

The authors adequately addressed the reviewer's comments and made a significant effort to make the paper ready for publication

The Excel figures style (10 to 14) can be improved in the final stage

good